# Emotional Components of Pain Perception in Borderline Personality Disorder and Major Depression—A Repetitive Peripheral Magnetic Stimulation (rPMS) Study

**DOI:** 10.3390/brainsci10120905

**Published:** 2020-11-24

**Authors:** Kathrin Malejko, André Huss, Carlos Schönfeldt-Lecuona, Maren Braun, Heiko Graf

**Affiliations:** 1Department of Psychiatry and Psychotherapy III, University of Ulm, 89075 Ulm, Germany; carlos.schoenfeldt@uni-ulm.de (C.S.-L.); marenbraun@gmx.de (M.B.); heiko.graf@uni-ulm.de (H.G.); 2Department of Neurology, University of Ulm, 89081 Ulm, Germany; andre.huss@uni-ulm.de

**Keywords:** repetitive peripheral magnetic stimulation, rPMS, pain, borderline personality disorder, depression

## Abstract

Various studies suggested alterations in pain perception in psychiatric disorders, such as borderline personality disorder (BPD) and major depression (MD). We previously investigated affective components of pain perception in BPD compared to healthy controls (HC) by increasing aversive stimulus intensities using repetitive peripheral magnetic stimulation (rPMS) and observed alterations in emotional rather than somatosensory components in BPD. However, conclusions on disorder specific alterations in these components of pain perception are often limited due to comorbid depression and medication in BPD. Here, we compared 10 patients with BPD and comorbid MD, 12 patients with MD without BPD, and 12 HC. We applied unpleasant somatosensory stimuli with increasing intensities by rPMS and assessed pain threshold (PT), cutaneous sensation, emotional valence, and arousal by a Self-Assessments Manikins scale. PTs in BPD were significantly higher compared to HC. The somatosensory discrimination of stimulus intensities did not differ between groups. Though elevated rPMS intensities led to increased subjective aversion and arousal in MD and HC, these emotional responses among intensity levels remained unchanged in BPD. Our data give further evidence for disorder-specific alterations in emotional components of pain perception in BPD with an absent emotional modulation among varying aversive intensity levels.

## 1. Introduction

Pain is considered to be an unpleasant emotional and sensory experience accompanied by potential or actual tissue damage. Apart from the somatosensory pathway, affective and cognitive components crucially modulate individual pain experience and perception [1]. Various studies have assumed clinically relevant alterations in pain perception in several psychiatric disorders and, in particular, in borderline personality disorder (BPD) [2,3,4,5,6] and major depressive disorder (MD) [7,8,9,10,11,12]. However, evidence on disorder specific alterations in pain perception in these two disorders is ambiguous depending on the different components of pain perception that are investigated and due to comorbidities, medication, and varying stimulus modalities.

Thus, previous studies investigating the experience of pain in MD revealed inconsistent findings. Elevated pain thresholds (PT) and attenuated pain sensitivity in MD were observed relative to healthy controls (HC) [7,9,10,12], particularly when pressure, thermal, or electrical stimuli were applied to the skin [8,11]. In contrast, one study demonstrated hyperalgesia to ischemic muscle pain in MD compared to HC [8]. Of note, most of these studies focused on somatosensory components of pain perception, whereas reliable evidence regarding alterations in emotional and cognitive components of pain perception in MD remains scarce.

In BPD, one core symptom is non-suicidal self-injury (NSSI) [13], which often manifests as cutting or burning, which is thought to regulate and relieve aversive inner tension [14,15]. Here, BPD patients frequently report hypo- or analgesia during NSSI [16]. Accordingly, various studies on pain perception observed reduced pain sensitivity in BPD compared to HC [2,3,4,5,6]. Whereas basic somatosensory stimulus perception and processing in BPD is thought to be unaffected, alterations in pain experience due to differences in affective or cognitive components of pain seem plausible [3,5,17]. We previously investigated affective components of pain perception in BPD compared to HC by parametrically increasing aversive stimulus intensities using repetitive peripheral magnetic stimulation (rPMS) [3]. The capability to discriminate different stimulus intensities did not differ between BPD and HC, but an elevation in levels of subjective aversion and arousal corresponding to intensity levels was solely observed in HC but not in BPD. These observations suggest preponderant alterations in emotional rather than somatosensory processes of pain perception in BPD, however, the specificity of these results was limited due to comorbid depression and/or medication in the investigated sample of BPD.

Based on those remaining and unanswered issues, we investigated a cohort of patients with BPD with comorbid MD, a sample of patients with MD without BPD, and HC to account for effects of depression and medication on emotional experience of pain in BPD. To warrant comparability with our previous findings, we applied unpleasant electrical stimuli with increasing stimulus intensities by rPMS and assessed participants’ (i) cutaneous sensation, (ii) emotional valence, and (iii) arousal using a Self-Assessments Manikins (SAM) scale. We assumed alterations regarding pain thresholds as well as emotional valence and arousal level during aversive stimulation specifically in BPD and intended to disentangle effects by comparisons with a clinical group diagnosed with MD without BPD with similar depressive symptoms and under medication.

## 2. Materials and Methods

### 2.1. Subjects

To account for gender differences and to minimize sample heterogeneity, we analyzed 34 females aged 18–55 years. Of those, 10 patients were diagnosed with BPD and comorbid MD and 12 patients with MD without BPD. A total of 12 healthy controls (HC) were investigated by a physician and served as a control group with no current or lifetime psychiatric diagnoses. Participants in the clinical groups and HC were matched for the highest degree of education and age. Participants were recruited from the inpatient units of the Department of Psychiatry and Psychotherapy III of the University Hospital Ulm. All patients in the MD- and 8 patients of the BPD-group took antidepressant medication, mainly selective serotonin reuptake inhibitors (SSRIs) or serotonin and norepinephrine reuptake inhibitors (SNRIs) (see Appendix A). Antidepressant medication was not interrupted but held stable for four weeks prior to the measurements. All participants were right-handed according to the Edinburgh Handedness Inventory. Participants with any severe medical disorder, epilepsy, current substance use disorder, and psychotic disorders were excluded from the study. All participants gave written informed consent prior to the study that was approved by the local ethical committee of Ulm University (Ethical approval code 52/09) and conducted in accordance with the Declaration of Helsinki. 

### 2.2. Psychometric Measurements

All participants were screened by using the Structured Clinical Interview for DSM-IV (SCID-I and -II [18]), and clinical diagnoses of patients with MD and BPD were verified by one of the study psychologists or physicians. Current depressive symptoms were assessed by using the Beck Depression Inventory (second edition, BDI-II [19]) in its German version [20]. NSSI was assessed by the German version of the Modified Ottawa/Ulm Self-injury Inventory (MOUSI) [21]. All patients in the BPD group committed NSSI at least once per week during the preceding 6 months. Psychological and somatoform dissociative features were assessed before and after rPMS procedures by the Dissociation Tension Scale-acute (DSS-acute) [22,23]. The DSS-acute is a self-rating instrument on a 10-level Likert scale (0–9). Ten items refer to psychological phenomena of dissociation, nine items include physical characteristics of dissociation, and two items describe borderline-specific symptoms. The total value is calculated from the 21 items and divided by the number of items. The presence of dissociative symptoms is assumed for patients who achieved higher values than 1.57 (unpublished cut-off value) [23].

### 2.3. Study Design

We examined pain thresholds (PT) on two consecutive days (T_1_&T_2_) while there were no significant differences between T_1_&T_2_ for all groups (PT: *p* = 0.09). We computed mean values to account for intraindividual variations in pain perception [24,25]. For rPMS procedures, all participants were seated in a comfortable chair and wore earplugs as well as headphones to reduce acoustic artefacts from magnetic impulses. The non-dominant arm rested extended on a table that was placed in front of the chair. With a pillow underneath, the palm pointed upwards in the direction of the ceiling. A circular parabolic coil (MMC-140 MacVenture, 140 mm, 33 kT/s^−1^) was placed in the palm with the handle pointing towards the opposite direction of the proband’s arm. The stimulator was a MagPro-X100 (2 Tesla) that was used to evoke an aversive sensation. An individual baseline of PT was defined anew before rPMS was applied in different intensities with a frequency of 25 Hz for 1 s. Intertrial interval (ITI) was 15 s. The starting point was at 10% of the stimulator’s maximum output intensity and with each step, stimulation intensity was increased by 10%. Immediately after each burst, participants were asked to evaluate its unpleasantness. Once the applied burst was described as unpleasant and almost painful, it was defined as one’s individual PT. When 100% of the maximum output was reached but not described as painful, this level was used as the reference baseline. After establishing one’s individual PT prior to each session, 50 bursts (25 Hz, 1 s) of rPMS with an ITI of 15 s were randomly delivered at five different intensities. Those five different intensities were selected as follows: the intensity of the burst that was described as painful (PT level) was set as a value of 5; all other levels of intensities (subthreshold values: 4–1) were applied in decreasing steps of 10%. The local aversive rPMS stimuli were evaluated immediately after each burst through Self-Assessments Manikins (SAM) [26] with three visual analogue scales (range: 1–9) representing the dimensions “cutaneous sensation” (scale 1; from “no pain at all” = 1 to “very painful” = 9), “emotional valence” (scale 2; from “pleasant” = 1 to “very unpleasant” = 9), and “level of arousal” (scale 3; from “I feel very calm” = 1 to “I feel an unbearable tension” = 9).

### 2.4. Data Analysis

Datasets were analyzed for normal distribution by a Shapiro–Wilk test, and respective statistical tests were chosen based on its outcome. Accordingly, descriptive statistics provided median values with 25% and 75% percentile or mean values with standard error of mean (SEM), respectively. Inter- and intra-group comparisons were performed by two-way analysis of variance (ANOVA) with Tukey’s multiple comparison test. All statistical tests were carried out by using GraphPad Prism 8 software (GraphPad Software Inc., La Jolla, CA, USA). A *p*-value ≤ 0.05 was considered as statistically significant.

## 3. Results

### 3.1. Demographic and Behavioral Data

A total of 12 patients with MD (*M_age_* = 31.8 (*SD* = 10.0)), 10 patients with BPD and comorbid MD (*M_age_* = 31.2 (*SD* = 8.1)) and 12 HCs (*M_age_* = 30.0 (*SD* = 4.4)) completed the study protocol and served for final data analysis. In line with the clinical diagnosis and the high comorbidity of MD in BPD [27], BDI scores indicated a moderate or severe major depression in both the MD and the BPD groups. Of note, BDI scores did not significantly differ between the BPD and the MD groups (*t* = 1.92; *p* = 0.069).

Regarding dissociative symptoms, we observed significantly higher DSS scores before than immediately after rPMS in the BPD group (*p* = 0.032, *t* = 2.65), while DSS scores between pre- and post-rPMS did not differ in the MD group (*p* = 0.703, *t* = 0.39) or between MD and BPD (pre-rPMS: *p* = 0.278, *t* = 1.11; post-rPMS: *p* = 0.576, *t* = 0.56) (see Appendix A).

### 3.2. rPMS Data

#### Pain Thresholds (PT)

An ANOVA revealed significant group-by-PTs interaction (*F* = 6.85, *p* = 0.004). PTs (in %) were significantly higher in BPD than in HC (*p* = 0.002), while there were no significant differences in PTs between BPD and MD (*p* = 0.215) or MD compared to HC (*p* = 0.130; see Figure 1).

### 3.3. Pain Perception

#### 3.3.1. Cutaneous Sensation (SAM Scale 1)

*Between groups:* No significant differences were observed between groups regarding the rating of cutaneous sensation for all intensity levels (see Figure 1).

*Within group:* We observed significant differences between cutaneous sensation for almost all intensity levels within all groups (see Table 1), indicating a similar capability to discriminate different and increasing levels of stimulus intensities.

#### 3.3.2. Emotional Valence (SAM Scale 2)

*Between groups:* Regarding the emotional valence rating, significantly higher values were found in BPD compared to MD (*p* = 0.010) as well as compared to HC (*p* = 0.031) at intensity level 1, while at intensity level 5, corresponding emotional valence was significantly lower in BPD compared to HC (*p* = 0.016) (see Figure 1).

*Within group:* No significant differences regarding emotional valence with increasing stimulus intensities were observed for BPD, while an increase in stimulus intensity was accompanied by elevated levels of aversion in patients with MD and HC (see Table 1).

#### 3.3.3. Arousal (SAM Scale 3)

*Between groups:* Regarding levels of subjective arousal, no significant differences were observed between groups for all intensity levels (see Figure 1).

*Within group:* No significant differences were observed within the BPD-group, while HC showed significant and patients with MD trendwise differences in their level of arousal at several intensity levels. Thus, HC and patients with MD revealed higher levels of arousal with higher levels of stimulation intensity (see Table 1).

## 4. Discussion

Here, we show patients with BPD and comorbid MD, patients with MD without BPD, and HC under different levels of unpleasant somatosensory rPMS-stimuli to elucidate previous reports regarding alterations of affective components of pain perception in BPD. By including two clinical samples in one study, we aimed to correct for potential confounding factors arising from MD and to disentangle disorder-specific alterations in pain processing in BPD. Psychometric measures revealed comparable depressive symptoms in both BPD with comorbid MD and MD without BPD. During rPMS, increasing aversive stimulus levels were similarly discriminated by all participants (SAM scale 1). While increasing rPMS stimulation intensity led to elevated subjective aversion and arousal levels in MD and HC, subjective emotional reactions were not modulated by unpleasant stimulus intensities in BPD (SAM scale 2 and 3).

In line with previous observations [4,6], we observed significantly higher PTs in BPD compared to HC, referring to a reduced pain sensitivity. Albeit not statistically significant, it is of note that we found a trend to higher PTs in BPD compared to MD and a similar trend towards increased PTs in MD compared to HC. However, particularly the lack of significant differences in PTs between BPD and MD may imply that PTs per se may not sufficiently differentiate between these disorders. In addition, the observation of significantly elevated PTs in BPD is not enough to conclude a generalized somatosensory deficit in BPD [28]. This is supported by the results of an unimpaired sensory discrimination of increasing somatosensory aversive stimuli intensities compared to MD and HC (SAM scale 1). In line with this observation, a recent neuroimaging study observed similar sensory stimulus intensity encoding neural activation within brain regions related to neural pain processing in BPD compared to HC [17].

We observed significant group differences regarding the assessment of emotional valence and arousal according to increasing stimulus intensities applied by rPMS. Whereas subjective emotional valence and arousal increased with elevating aversive stimulus intensities in both MD and HC, this intensity related modulation of emotional valence and arousal was not evident in BPD. In particular, we observed an increased subjective emotional valence and arousal in BPD compared to HC at low aversive rPMS stimulus levels and lower emotional valence and arousal in BPD compared to HC at high rPMS stimulus levels. This pattern led to the assumption of a disorder specific alteration in affective appraisal of pain in BPD that also differentiates patients with BPD and comorbid MD from MD without BPD compared to HC. Thus, our data strengthen recent findings from neuroimaging studies which demonstrated an altered neural pain processing in BPD with increased neural activation within the prefrontal cortex but attenuated activation of the anterior cingulate cortex and limbic regions [4,29,30]. Accordingly, this pattern was interpreted as a neuroanatomical proxy of an anti-nociceptive mechanism through downregulation of the emotional aspects of pain processing by increased top-down regulation. Alternatively, the lack of variation in emotional valence with changing intensity levels in BPD may be due to altered emotional perception or the ability to differentiate emotions, which is clinically often observed.

However, it has to be mentioned that we were only able to investigate a comparably small sample size that compromises the generalizability of our results. The small sample size may also account for the lack of statistically significant differences in PTs between MD and HC. Nevertheless, the trend of higher PTs in MD compared to HC found in our study is in line with previous findings [7,9,10,12]. Moreover, considering the antinociceptive effects of antidepressants, especially SSRIs [31,32], it is of note that we investigated patients with BPD and MD under medication that may potentially confound our results. However, alterations in affective-motivational components of pain in BPD have also been observed in absence of medication [4,29,30,33] which support our findings. Another shortcoming that needs be mentioned is the fact that we cannot exclude trial-by-trial time effects on subjective evaluation. To correct for individual fluctuations regarding the subjective evaluation of intensity, unpleasantness, and arousal over time, we performed these measures on two different (consecutive) days and could not detect significant differences in these inter-day comparisons.

## 5. Conclusions

We investigated patients with BPD and comorbid MD and patients with MD without BPD and compared them to HC to elucidate BPD-specific alterations in pain perception by controlling for potential confounds owing to comorbidity and medication. Increasing levels of unpleasant stimuli were applied via rPMS, and we assessed participants’ PT, subjective cutaneous sensation, emotional valence, and arousal level by SAM scales. Our study supports previous results of elevated pain thresholds in BPD compared to HC. In addition, we did not find any significant differences compared to MD, which could indicate that this finding may not be a disorder specific alteration. During rPMS, we found no significant differences between groups regarding the somatosensory discrimination of increasing stimulus intensities levels (SAM scale 1). Increasing levels of stimuli intensities led to elevated emotional valence and arousal level only in the MD and the HC groups, whereas, in BPD patients, responses remained unchanged among different intensity levels. BPD patients did not show a modulation in their emotional reaction to increasing intensity levels of unpleasant somatosensory stimulation. Thus, we provide further evidence regarding disorder-specific alterations in emotional components of pain perception in BPD with an absent emotional modulation among varying aversive intensity levels.

## Figures and Tables

**Figure 1 brainsci-10-00905-f001:**
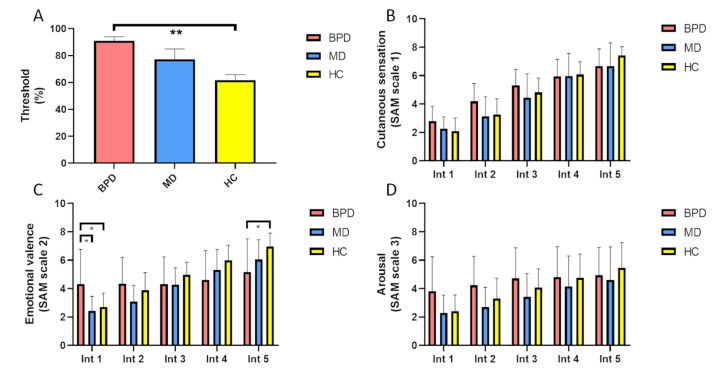
(**A**) Pain threshold (PT) of patients with borderline personality disorder (BPD) and comorbid major depression, patients with major depressive disorder (MD) without BPD, and healthy controls (HC). (**B**–**D**) Differences between groups regarding their subjective rating of Self-Assessments Manikins (SAM) scale 1, 2, and 3 for all levels of increasing stimulus intensities. * = *p* < 0.05; ** = *p* < 0.01; Int = intensity level.

**Table 1 brainsci-10-00905-t001:** Differences between intensity levels within group of patients with BPD and comorbid major depression, patients with MD without BPD, and HC.

	BPD (*n* = 10)			MD (*n* = 12)			HC (*n* = 12)		
SAM Scale	1	2	3	1	2	3	1	2	3
Int. 1 vs. Int. 2	ns	ns	ns	ns	ns	ns	ns	ns	ns
Int. 1 vs. Int. 3	****	ns	ns	***	*	ns	****	**	ns
Int. 1 vs. Int. 4	****	ns	ns	****	****	ns	****	****	*
Int. 1 vs. Int. 5	****	ns	ns	****	****	*	****	****	***
Int. 2 vs. Int. 3	ns	ns	ns	ns	ns	ns	*	ns	ns
Int. 2 vs. Int. 4	*	ns	ns	****	**	ns	****	**	ns
Int. 2 vs. Int. 5	****	ns	ns	****	****	ns	****	****	*
Int. 3 vs. Int. 4	ns	ns	ns	*	ns	ns	ns	ns	ns
Int. 3 vs. Int. 5	ns	ns	ns	***	*	ns	****	*	ns
Int. 4 vs. Int. 5	ns	ns	ns	ns	ns	ns	ns	ns	ns

Cutaneous sensation = SAM scale 1; emotional valence = SAM scale 2; arousal = SAM scale 3; Int. = intensity level; ns = not significant; * = *p* < 0.05; ** = *p* < 0.01; *** = *p* <0.001; **** = *p* < 0.0001.

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
