# Peer review of "Emotional Components of Pain Perception in Borderline Personality Disorder and Major Depression—A Repetitive Peripheral Magnetic Stimulation (rPMS) Study"

_brainsci, 2020, doi:10.3390/brainsci10120905_

Round 1
Reviewer 1 Report
This is an interesting study that aims at disentangling depression and emotion regulation disturbances in Borderline Personality Disorder patients comparing a group of patients with depression with BPD to depression without BPD and to a group of healthy control subjects using externally induced pain stimuli.
The authors found no evidence for a lowered pain threshold in both patient groups but an increased pain threshold in BPD compared to HC with the "pure" MD patients in the middle. Interestingly, stimulus intensity had no influence on perceived arousal (not significan) and valence in BPD. This could point to a reduced self-awareness or emotional awareness in BPD (my interpretation), although the authors explain it more within the framework of pain perception and processing.
Overall, despite the quite low number of subjects, it is an interesting study with a clean design and a clean paradigm.
I have only some questions and comments:
Why did the authors limit their sample to female patients? This should at least be explained...
Were other personality disorders detected or excluded in the patient samples?
How were healthy people screened?
Was there any change in evaluation of intensity, unpleasantness or arousal across time (i.e. from the start of each experiment through the end)? And was there any difference in such effects between the groups?
Discussion:
The authors argue that the lack of variation of emotional valence with changing intensity levels in BPD is due to altered neural pain processing in BPD. What about altered emotion perception or ability to differentiate emotions and their change in BPD as an alternative explanation?
Some minor (linguistic) comments:
- P. 2, lines 66-67: correct to "participants' i) cutaneous sensation, ii emotional valence and iii) arousal using a Self-Asstessment Manikins (SAM) scale."
- P. 7 (?), line 184: suggestion: "By including two clinical samples in one study we aimed at ..."
- P. 7, line 192: "we observed significantly higher..."
- P. 7, line 208: something is missing in "This pattern of assume..."
Reviewer 2 Report
This manuscript represents an interesting and relevant approach to disentangle the effect of MD from BPD, which often co-occurs. The methodology seems to be of good quality. The major problem is the small sample size, and the conclusions drawn in relation to this. Also, I think the Discussion needs more work. Here are my comments:
- I was not able to see the supplementary Table S1. In this I would have liked to see:
- Parameters BSS and BDI at baseline between groups
- Which medication the participants with BPD and MD were taking (particularly since SSRI and SNRI may have analgesic effects, while other medications may not).
- Was NSSI an inclusion criterion for patients with BPD? NSSI at least once a week for the last 6 months for all seems quite prevalent.
- Results Line 139-142. "...and between groups (pre-rPMS: p=0.278, t=1.11; 141 post-rPMS: p=0.576, t=0.56) (see supplementary material table S1)".
Does this refer to betweem MD and BPD (not clear)?
- Please provide reason for the participants who did not complete the study protocol.
- Page 3, line 112-113 "Once the applied burst was described as almost painful..."
Why are the PT:s only "almost painful"?
- Figure 1 SAM-1 It doesn't seem that the pain rating increases that much for the BPD group between 3-4-5? Even if they can discriminate between the sensations, if they are not experienced as more painful, it might not be so surprising that the levels for SAM-2 and SAM-3 did not increase during the stimulations 3-4-5 for BPD? Please elaborate.
- Please provide the unit for the pain threshold in Figure 1 and in the text (Results).
- Name the included figures in Figure 1, for example: 1a, 1b etc., and provide figure legends for each of them.
- Discussion: Line 195-196."However, particularly the lack of significant differences in PTs between BPD and MD led to the assumption that PTs per se may not sufficiently differentiate between these disorders." This is a strong conclusion from a study with a very small sample size.
- Discussion line 197-198: "In addition, the observation of significantly elevated PTs in BPD is not eligibable to conclude generalized somatosensory deficits. "
This is an unclear sentence.
- Eligible in the sentence above is not correctly spelled
- Line 204-206: "Whereas subjective emotional valence and arousal increased with elevating aversive stimulus intensities in both, MD and HC, this intensity related modulation of emotional valence and arousal was not evident in BPD."
Remove "," between both and MD. Punctuations that are not correct make the manuscript hard to read.
- 206-207 "In particular, we observed an increased subjective emotional valence and arousal in BPD compared to HC at low and high aversive rPMS stimulus levels. "
This does not seem to be right, didn't you observe higher emotional valence in BPD compared to HC at low rPMS stimulus levels, and lower emotional valence and arousal in BPD compared to HC at high rPMS stimulus levels.
- 208: "This pattern of assume a disorder specific..."
This is a strange sentence.
- Line 222-223 "Though, alterations in affective-motivational components of pain in BPD were also observedin the absence of medication [44,29,30,33] and support our findings."
Unclear what you refer to. Rather use: "...in BPD have also been observed in absence.... which support our findings".
- Conclusion.Lines:229-230. "Our study supports previous results of elevated pain thresholds in BPD compared to HC, but may not serve as disorder specific alterations since they did not differ significantly from MD."
Again, this is a very strong conclusion for a study this sample size.
